# ANTIFAKEPROMPT: PROMPT-TUNED VISION-LANGUAGE MODELS ARE FAKE IMAGE DETECTORS

**You-Ming Chang**[1][*]   **Chen Yeh**[1][*]   **Wei-Chen Chiu**[1]   **Ning Yu**[2]
[1]National Yang Ming Chiao Tung University   [2]Netflix Eyeline Studios
{thisismiiiing.11, denny3388.cs11}@nycu.edu.tw
walon@nctu.edu.tw, ningyu.hust@gmail.com

## ABSTRACT

Deep generative models can create remarkably photorealistic fake images while raising concerns about misinformation and copyright infringement, known as deepfake threats. Deepfake detection technique is developed to distinguish between real and fake images, where the existing methods typically learn classifiers in the image domain or various feature domains. However, the generalizability of deepfake detection against emerging and more advanced generative models remains challenging. In this paper, being inspired by the zero-shot advantages of Vision-Language Models (VLMs), we propose a novel approach called AntifakePrompt, using VLMs (e.g., InstructBLIP) and prompt tuning techniques to improve the deepfake detection accuracy over unseen data. We formulate deepfake detection as a visual question answering problem, and tune soft prompts for InstructBLIP to answer the real/fake information of a query image. We conduct full-spectrum experiments on datasets from a diversity of 3 held-in and 20 held-out generative models, covering modern text-to-image generation, image editing and adversarial image attacks. These testing datasets provide useful benchmarks in the realm of deepfake detection for further research. Moreover, results demonstrate that (1) the deepfake detection accuracy can be significantly and consistently improved (from 71.06% to 92.11%, in average accuracy over unseen domains) using pretrained vision-language models with prompt tuning; (2) our superior performance is at less cost of training data and trainable parameters, resulting in an effective and efficient solution for deepfake detection.

## 1 INTRODUCTION

In recent years, we have witnessed the magic leap upon the development of deep generative models, where the cutting-edge models such as Stable Diffusion (Rombach et al., 2022), DALLE-2 (Ramesh et al., 2022), and DALLE-3 (OpenAI, 2023) have become capable of producing high-quality images, ranging from beautiful artworks to incredibly realistic images.

However, the progress of such image synthesis technique, which is called "deepfake", poses real threats to our society, as some realistic fake images could be produced to deceive people and spread false information. For example, images of the war in Ukraine could be generated with false or misleading information and may be used for propaganda[1]. More concerningly, some of these created images may be falsely claimed as works of photographers or artists, potentially leading to copyright infringements and their misuse in commercial contexts. As BBC reported, some fake artworks generated by text-to-image generation models won first place in an art competition, harming the fairness of the contest[2]. To protect against these threats arising from deepfake content, the use of effective deepfake detection techniques becomes crucial. These techniques help distinguish real content from manipulated images, serving as a vital defense against deception and safeguarding intellectual property rights in the digital era.

---

[*]Both authors contribute equally

[1]https://techcrunch.com/2022/08/12/
a-startup-wants-to-democratize-the-tech-behind-dall-e-2-consequences-be-damned/

[2]https://www.bbc.com/news/technology-62788725

One straightforward deepfake detection prototype is to train a classifier to distinguish between real and fake images (Wang et al., 2020; Guarnera et al., 2023; Yu et al., 2019). However, along with the rapid development of generative models, this approach often struggles with overfitting, thus leading to poor performance on unseen data from emerging generators. To overcome this limitation, researchers are exploring more general features in deepfake images, such as frequency maps of the images (Zhang et al., 2019b). Additionally, some innovative methods are not only based on visual features. For example, DE-FAKE (Sha et al., 2022) harnesses the power of large language models and trains a classifier that conditions on both visual and textual information.

Despite years of development in deepfake detection techniques, challenges persist. First, most previous works such as (Wang et al., 2020) have concentrated on Generative Adversarial Networks (GANs), which may not effectively address the latest diffusion-based generative models including Stable Diffusion (Rombach et al., 2022) (SD), Stable Diffusion-XL (Podell et al., 2023) (SDXL), DALLE-2 (Ramesh et al., 2022), DeepFloyd IF (StabilityAI, 2023). Second, generalizability remains a significant challenge. Classifiers trained on images generated by one model tend to perform poorly when being tested on images from different generative models, especially from more emerging and advanced ones.

To address the aforementioned challenges and to utilize the strong generality of LLMs, we have harnessed the zero-shot capabilities of pretrained vision-language models (Li et al., 2022; 2023; Zhu et al., 2023; Liu et al., 2023b; Dai et al., 2023) to capture more general instruction-aware features from images, enhancing the transferability of our proposed deepfake detector, **AntifakePrompt**. We formulate the deepfake detection problem as a Visual Question Answering (VQA) task, asking the model with the question "Is this photo real?", to tackle this challenge. However, directly asking questions for a pretrained VLM may not lead to effective answers, considering either the query images or questions are unseen during VLM training. We therefore use prompt tuning to boost the performance. Without loss of generality, we build implementation on the recent state-of-the-art VLM, InstructBLIP (Dai et al., 2023). Specifically, we insert a "pseudo-word" into the prompt and optimize the corresponding word embedding in the model for correctly answering "Yes" and "No" to the question for real and fake images on training data, respectively. AntifakePrompt not only significantly reduces training costs but also substantially improves performance on both held-in and held-out testing datasets from a full spectrum of generative models. From the perspective of instruction tuning, we realized that there are many good answers, all waiting for a good question. In summary, our paper makes the following key contributions:

1. We pioneer to leverage pretrained vision-language models to solve the deepfake detection problem. We are the first to formulate the problem as a VQA scenario, asking the model to distinguish between real and fake images. Additionally, we employ soft prompt tuning techniques to optimize for the most effective question to the VLMs, and leverage their zero-shot generalizability on unseen data produced by held-out generative models.

2. We meticulously curate 23 testing datasets, encompassing six frequently encountered deepfake scenarios and diverse real images to evaluate the generalizability and effectiveness of baseline methods and our proposed detector, AntifakePrompt. These datasets can also serve as benchmarks for future advancements in deepfake detection.

3. Our detector consistently outperforms the recent baseline methods proposed in (Ricker et al., 2022; He et al., 2016; Liu et al., 2024; Wang et al., 2020; Sha et al., 2022; Wang et al., 2023b; Wu et al., 2023; Le & Woo, 2023) over held-out datasets generated by a full spectrum of generator categories. Our superior performance and generalizability benefit from the nature of pretrained VLMs, and at less cost of training data and trainable parameters.

## 2 RELATED WORK

### 2.1 VISUAL GENERATIVE MODELS

The recent advance of deep generative models can be broadly categorized into two main types: Generative-Adversarial-Networks-based (GAN-based) models and diffusion-based models. Within the realm of GAN-based model, notable progress has been made. Starting from GAN (Goodfellow et al., 2014), SA-GAN (Zhang et al., 2019a) and BigGAN (Brock et al., 2018) contributed to the

enhancement of training stability and the generation of diverse images with higher resolution. Subsequently, StyleGAN (Karras et al., 2019) and its successors (Karras et al., 2020; 2021) have allowed for finer control over the stylistic attributes of the generated images while maintaining high image quality. Building upon StyleGAN-3 and ProjectGAN (Sauer et al., 2021), StyleGAN-XL (Sauer et al., 2022) (SGXL) is able to generate $1024 \times 1024$ images with even lower Fréchet Inception Distance (FID) scores and higher Inception Scores (IS), w.r.t. all its predecessors.

In regard to the diffusion-based models, starting from DDPM (Ho et al., 2020), DDIM (Song et al., 2020) speeds up the generating process by relaxing the constraint of Markov Chain towards forward and backward processes. Latent Diffusion (Rombach et al., 2021) and Stable Diffusion (Rombach et al., 2022) (SD) further shift the diffusion process to latent space, granting user controls over the models; thus, it flexibly enables the text-to-image generation through diffusion-based models. Building upon this foundation, several seccessors (e.g., SDXL (Podell et al., 2023), DeepFloyd IF (StabilityAI, 2023), , DALLE-2 (Ramesh et al., 2022), and DALLE-3 (OpenAI, 2023)) further refine the text comprehension capabilities of diffusion-based models, enabling them to create images that better align with input texts.

Apart from text-to-image generation, image editing tasks, such as inpainting and super resolution, are also widely-used applications of generative models. Notably, (Suvorov et al., 2022; Rombach et al., 2022) have demonstrated exceptional performances in the domain of image inpainting, while (Rombach et al., 2022; Chen et al., 2021) are known for their remarkable performances in image super resolution.

Without the loss of representativeness, we select a diverse set of generative models (namely SD2 , SD3 (Rombach et al., 2022), SDXL, IF, DALLE-2, DALLE-3, SGXL, ControlNet (Zhang & Agrawala, 2023), LaMa (Suvorov et al., 2022), LIIF (Chen et al., 2021), SD2 inpainting model, SD2 super resolution model), GLIDE (Nichol et al., 2021), Playground (Li et al., 2024) to cover the full spectrum of generation tasks, and generate corresponding fake images for conducting our experiments.

## 2.2 DEEPFAKE DETECTION METHODS

Recent advances in detection methods have focused on training detectors capable of identifying artifacts specific to certain types of generative models. For example, (Wang et al., 2020; Nataraj et al., 2019; Yu et al., 2019; Ricker et al., 2022; Wang et al., 2023b; Wu et al., 2023; Ma et al., 2023; Lorenz et al., 2023) leverage artifacts from synthesized images generated by GANs or diffusion models, (Zhang et al., 2019b; Giudice et al., 2021; He et al., 2021) concentrate on artifacts in the frequency domain, and (Le & Woo, 2023) also study the detection of low-quality or low-resolution fake images. These methods have reported outstanding performance on images generated by the seen models, but they often suffer from significant drops in performance when being applied to unseen datasets. Therefore, we aim to propose a general detector that can demonstrate exceptional performance on both held-in and held-out datasets.

## 2.3 VISION-LANGUAGE MODELS AND VISUAL QUESTION ANSWERING

With the impressive success of Large Language Models (LLM), recent studies work on Vision-Language Models (VLMs) to improve multimodal comprehension and generation through utilizing the strong generalizability of LLMs. These models takes advantage of cross-modal transfer, allowing knowledge to be shared between language and multimodal domains. BLIP-2 (Li et al., 2023) employing a Flan-T5 (Chung et al., 2022) with a Q-former to efficiently align the visual features with language model. InstructBLIP (Dai et al., 2023) also utilizes the pretrained visual encoder and Q-former from BLIP-2, with Vicuna/Flan-T5 as pretrained LLM, but performs instruction tuning on Q-former using a variety of vision-language tasks and datasets. LLaVA (Liu et al., 2023b) projects the output of a visual encoder as input to a LLaMA/Vinuca LLM with a linear layer, and finetunes the LLM on vision-language conversational data generated by GPT-4 (OpenAI, 2023) and ChatGPT. CogVLM (Wang et al., 2023a) finetunes extra modules added in each layer of the transformer-based language model (i.e., Vicuna7B (Chiang et al., 2023)), where these extra modules named "visual expert" are composed of the QKV matrices and MLP layers identical to those in the original transformer, to enable better alignment of language embeddings with respect to the visual features extracted by vision transformer.

Among the vision-language tasks, visual question answering (VQA) problem is one of the most general and practical tasks because of its flexibility in terms of the questions. For training the models for VQA problems, lots of datasets have been proposed. VQAv2 (Goyal et al., 2017) and VizWiz (Gurari et al., 2018) collect images, questions, as well as the corresponding answers for studying visual understanding. OKVQA (Marino et al., 2019) and A-OKVQA (Schwenk et al., 2022) propose visual question-answer pairs with external knowledge (e.g., Wikipedia). On top of that, the aforementioned VLMs can also be potential solutions, as they have strong multimodal comprehension and generality.

Given the remarkable multimodal capabilities of VLMs, we have harnessed their potential to address the deepfake detection challenge. Therefore, we formulate the deepfake detection method as a VQA problem to take advantage of the capabilities of these VLMs.

Furthermore, as (Chen et al., 2023; Zou et al., 2023; Deng et al., 2022; Zhang et al., 2022) concluded, prompt tuning offers an approach to enabling Langnuage Models (LMs) to better understand user-provided concepts, and improves the alignments between generated images and the input prompts when applied to text-to-image generative models (Wen et al., 2023; Gal et al., 2022). Inspired by these findings, we apply prompt tuning atop InstructBLIP to optimize an instruction that can more accurately describe the idea of differentiating real and fake images, resulting in better performance.

# 3 ANTIFAKEPROMPT

## 3.1 PROBLEM FORMULATION

In order to take advantage of the vision-language model, we formulate the deepfake detection problem as a visual question answering (VQA) problem. In this framework, the input consists of a query image $\mathbf{I}$ that needs to be classified as real or fake and a question prompt $q$. The prompt can be either a preset question (e.g., "Is this photo real?") or a tunable question that includes the pseudo-word $S_*$. The output of this framework corresponds to the answer texts $y$. While $y$ in principle can be any texts, we constrain it to two options: "Yes" and "No" during testing, aligning with the answer ground truth to the original binary classification problem. We choose the option with a higher probability from the VLM as the answer, where the model capability is evaluated by classification accuracy.

In summary, the deepfake detection task can be formulate as a VQA task, which is defined as:

$$\mathcal{M}(\mathbf{I}, q) \mapsto y \tag{1}$$

where $\mathcal{M}$ is an VLM and we adopt InstructBLIP for building our method, and the text output $y \in \{\text{"Yes"}, \text{"No"}\}$ corresponds to the binary results of deepfake detection.

## 3.2 PROMPT TUNING ON INSTRUCTBLIP

As discussed in (Dai et al., 2023), the prompt plays an essential role in VQA problem, and asking the preset question leads to ineffective performance on unseen data. Therefore, we employ soft prompt tuning on InstructBLIPfollowing the procedure below. The reason why we choose InstructBLIP as our backbone model is described in paragraph 1 of section 4.2.

Within InstructBLIP, two components receive the prompt as input: Q-Former and the Large Language Model (LLM). As shown in fig. 1, the prompt first gets tokenized and embedded, and then is fed into Q-Former and the LLM in parallel. We introduce a pseudo-word $S_*$ into the prompt, which serves as the target for soft prompt tuning. Specifically, we adopt the question template, "Is this photo real?" and append the pseudo-word to the end of the prompt, resulting in the modified prompt $q_*$: "Is this photo real $S_*$?". As the prompt has been decided, we give the output label $\hat{y} = $ "Yes" for real images and $\hat{y} = $"No" for fake images in order to perform soft prompt tuning.

We freeze all parts of the model except for the word embedding $v$ of the pseudo-word $S_*$, which is randomly initialized. Then we optimize the word embedding $v_*$ of the pseudo-word over a training set of triplet $\{\mathbf{I}, q_*, \hat{y}\}$ with respect to the language modeling loss, expecting the VLM output $y$ to be the label $\hat{y}$. Hence, our optimization goal can be defined as:

$$\widetilde{S_*} = \arg\min_{S_*} \mathbb{E}_{(\mathbf{I}, \hat{y})} \mathcal{L}(\mathcal{M}(\mathbf{I}, \text{"Is this image real } S_*\text{"}), \hat{y}) \tag{2}$$

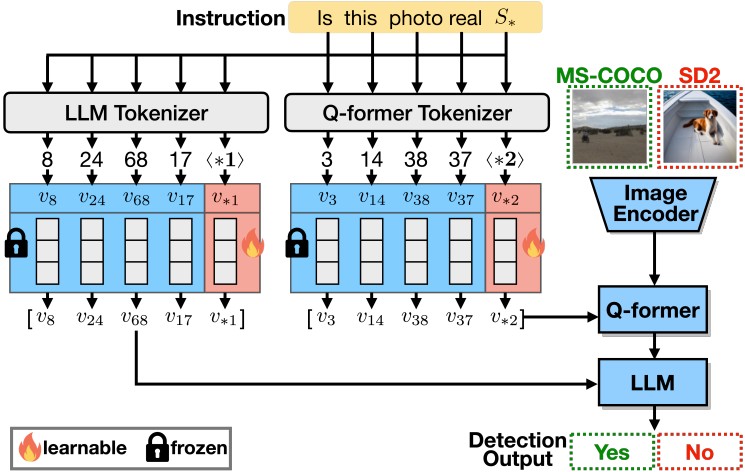

Figure 1: **Prompt tuning on InstructBLIP (Dai et al., 2023) for deepfake detector training**. An instruction containing a pseudo-word $S_*$ is first converted into tokens. These tokens are converted to continuous vector representations (the "embeddings", $v$). Then, the embedding vectors are fed into Q-former and LLM with the image features extracted by the image encoder. Finally, the embedding vectors $v_{*1}$ and $v_{*2}$ are optimized using language modeling loss, expecting the output to be "Yes" for real images and "No" for fake images.

where $\mathcal{L}$ is the language modeling loss function. Since we actually optimize the embedding $v_*$ for the pseudo-word $S_*$, with noting the concatenation of embeddings for the original prompt (i.e. "'Is this photo real") to be $v_p$, the equation can be rewritten as:

$$\widetilde{v_*} = \arg \min_{v_*} \mathbb{E}_{(\mathbf{I},\hat{y})} \mathcal{L}(\mathcal{M}(\mathbf{I}, [v_p, v_*]), \hat{y}) \tag{3}$$

As fig. 1 shows, it is crucial to highlight that the pseudo-word embedding fed into Q-Former $v_{*1}$ differs from that fed into the LLM $v_{*2}$, and we optimize these two embeddings independently. The dimensions of $v_{*1}$ and $v_{*2}$ are 768 and 4096 respectively, so the number of trainable parameters is 4864 in total. Compared to 23 million trainable parameters from ResNet-50 (He et al., 2016) of (Wang et al., 2020) and 11 million trainable parameters from ResNet-18 (He et al., 2016) of DE-FAKE (Sha et al., 2022), our method demonstrates superior cost-efficiency.

**Implementation details.** We use the LAVIS library[3] for implementation, training, and evaluation. To avoid the out-of-memory issue on small GPUs, we choose Vicuna-7B (Chiang et al., 2023), a decoder-only Transformer instruction-tuned from LLaMA (Touvron et al., 2023), as our LLM. During prompt tuning, we initialize the model from instruction-tuned checkpoint provided by LAVIS, and only finetune the word embedding of the pseudo-word while keeping all the other parts of the model frozen. All models are prompt-tuned with a maximum of 10 epochs. We use AdamW (Loshchilov & Hutter, 2017) optimizer with $\beta_1 = 0.9$ and $\beta_2 = 0.999$, batch size 6 and a weight decay 0.05. We initially set the learning rate to $10^{-8}$, and apply a cosine decay with a minimum learning rate of 0. All models are trained utilizing 4 NVIDIA RTX 3090 GPUs and completed within 10 hours. In terms of image preprocessing, all images are initially resized to have a length of 224 pixels on the shorter side while maintaining their original aspect ratio. During the training phase, random cropping is applied to achieve a final size of $224 \times 224$ pixels, while images are center cropped to a final size of $224 \times 224$ pixels in the testing phase.

## 4    EXPERIMENTS

To determine the optimal configuration for AntifakePrompt, we begin with a comparative analysis of pretrained VLMs, such as CogVLM, as a pilot study to select the backbone for AntifakePrompt.

---

[3] https://github.com/salesforce/LAVIS

Table 1: **Ablation Studies**: Each partition represents different ablation studies, namely "Position of $S_*$ in tuned prompt", "Prompt tuning for Q-former, LLM or both.", and "Number of training data" from left to right respectively. Entries are the accuracies of real and fake held-out datasets. The best performances of each ablation study are in **bold**.

| | S* Position Variants | | | Prompt Tuning for | | | Number of Training Data | | | | | | |
| --- | --- | --- | --- | --- | --- | --- | --- | --- | --- | --- | --- | --- | --- |
| | Replace | Prefix | Postfix | Only Q-former | Only LLM | Both | 90K | 120K | 150K | 180K | 15K | 1.5K | 0.15K |
| **COCO** | 95.13 | **95.80** | 95.37 | 93.50 | 95.10 | **95.37** | 89.90 | 94.30 | 95.37 | 96.53 | 92.60 | 92.83 | **99.07** |
| **Flickr** | 86.20 | 89.57 | **91.00** | **92.27** | 85.57 | 91.00 | 80.37 | 89.10 | 91.00 | 92.23 | 92.03 | 94.10 | **99.73** |
| **SD2** | 95.80 | 97.27 | **97.83** | **97.93** | 95.77 | 97.83 | **98.33** | 97.53 | 97.83 | 97.37 | 92.17 | 75.67 | 33.17 |
| **SDXL** | 93.60 | 96.47 | **97.27** | **98.17** | 91.73 | 97.27 | **98.20** | 96.60 | 97.27 | 96.67 | 90.20 | 68.20 | 16.73 |
| **IF** | 87.33 | 88.77 | **89.73** | 88.47 | 85.23 | **89.73** | **92.87** | 89.20 | 89.73 | 88.43 | 75.80 | 58.17 | 16.30 |
| **DALLE-2** | 99.17 | 97.90 | **99.57** | 99.53 | 98.73 | **99.57** | 98.90 | 98.27 | 99.57 | **99.63** | 97.27 | 79.77 | 36.93 |
| **SGXL** | 98.37 | 99.87 | **99.97** | 94.80 | 97.80 | **99.97** | 99.93 | 99.90 | **99.97** | 99.93 | 97.97 | 90.63 | 56.73 |
| **GLIDE** | 95.47 | **99.97** | 99.17 | 98.37 | 97.33 | **99.17** | 99.33 | **99.47** | 99.17 | 99.23 | 97.30 | 95.57 | 98.93 |
| **ControlNet** | **99.70** | 89.43 | 91.47 | 89.33 | 84.90 | **91.47** | **94.60** | 91.03 | 91.47 | 90.43 | 81.27 | 67.50 | 26.17 |
| **DF** | 93.03 | 94.17 | **97.90** | **100.00** | 93.30 | 97.90 | **99.27** | 95.13 | 97.90 | 94.63 | 91.93 | 67.53 | 24.20 |
| **DFDC** | 99.97 | **100.00** | **100.00** | **100.00** | 98.30 | **100.00** | **100.00** | **100.00** | **100.00** | **100.00** | 99.83 | 95.20 | 97.47 |
| **FF++** | 99.33 | **100.00** | 97.43 | 99.27 | **99.53** | 97.43 | **100.00** | **100.00** | 97.43 | **100.00** | 98.87 | 92.03 | 95.13 |
| **LaMa** | 33.40 | **40.33** | 39.03 | 37.67 | 31.37 | **39.03** | **49.80** | 41.03 | 39.03 | 36.87 | 34.73 | 30.80 | 8.90 |
| **SD2IP** | 78.63 | 84.67 | **85.20** | 77.50 | 77.77 | **85.20** | **89.63** | 85.13 | 85.20 | 84.27 | 73.93 | 67.57 | 40.53 |
| **LIIF** | 97.97 | **100.00** | 99.97 | 99.93 | 99.87 | **99.97** | 99.80 | 99.90 | **99.97** | 99.93 | 99.10 | 97.37 | 99.03 |
| **SD2SR** | 99.70 | 99.87 | **99.93** | 99.67 | 99.40 | **99.93** | **99.93** | 99.87 | 99.93 | 99.93 | 98.43 | 90.00 | 63.53 |
| **Adver.** | 90.43 | 93.53 | **96.70** | **97.53** | 86.03 | 96.70 | **98.40** | 97.87 | 96.70 | 94.83 | 84.07 | 29.57 | 3.97 |
| **Backdoor** | 86.00 | **93.13** | 93.00 | **97.97** | 83.47 | 93.00 | **96.40** | 96.00 | 93.00 | 91.50 | 82.57 | 25.93 | 2.40 |
| **Data Poisoning** | 86.63 | 87.43 | **91.57** | **95.23** | 83.50 | 91.57 | 94.97 | **95.37** | 91.57 | 90.57 | 74.53 | 11.87 | 1.43 |
| **Average** | 90.31 | 92.01 | **92.74** | 92.48 | 88.67 | **92.74** | **93.72** | 92.93 | 92.74 | 92.26 | 87.08 | 70.02 | 48.44 |

Subsequently, we conduct several ablation studies on a subset of testing datasets. Finally, we validate AntifakePrompt with the identified optimal configuration for further comparison with SOTA methods.

## 4.1 SETUP

**Real datasets.** We use Microsoft COCO (COCO) (Lin et al., 2014) dataset and Flickr30k (Young et al., 2014) dataset. In our work, we selected 90K images, with shorter sides greater than 224, from COCO dataset for the real images in the training dataset. Moreover, to assess the generalizability of our method over various real images, we additionally select 3K images from Flickr30k dataset to form a held-out testing dataset, adhering to the same criterion of image size.

**Fake datasets.** In order to evaluate the generalizability and robustness of our model to fake images from emerging and unseen generators, our testing datasets include fake images from 18 different generative models/datasets and 3 distinct attack scenarios, and each of the testing datasets comprises 3K images. We can mainly divide these images into six categories as follows: (1) **Text-to-images generation**: SD2, SD3, SDXL, DeepFloyd IF, DALLE-2, DALLE-3, Playground, SGXL, GLIDE, DiffusionDB (Wang et al., 2022b); (2) **Image stylization**: ControlNet; (3) **Image inpainting**: LaMa, SD2-Inpainting (SD2IP); (4) **Super Resolution**: LIIF, SD2-SuperResolution (SD2SR) (5) **Face Swap**: DeeperForesics (DF) (Jiang et al., 2020), DFDC (Dolhansky et al., 2019), Face-Forensics++ (FF++) (Rossler et al., 2019) (6) **Image attacks**: adversarial attack (Kim, 2020), backdoor attack (Li et al., 2021) and data poisoning attack (Geiping et al., 2020) (denoted as Adver., Backboor, and Data Poison., respectively). Please refer to appendix for the dataset generation details.

As for the fake images in the training dataset, it is important to note that we only include 30K images generated by each of SD3 (SD2 for ablation studies) and SD2IP, forming a total 150K training dataset. Empirical evidence demonstrates that AntifakePrompt, trained solely on these two fake datasets and the real images from COCO dataset, exhibits excellent performance on all the other datasets generated by held-out generative models.

**Baselines.** We compare our AntifakePrompt to several recent baseline models, i.e., **Ricker2022** (Ricker et al., 2022), **ResNet** (He et al., 2016), **FatFormer** (Liu et al., 2024), **Wang2020** (Wang et al., 2020), **DE-FAKE** (Sha et al., 2022), **DIRE** (Wang et al., 2023b), **LASTED** (Wu et al., 2023), **QAD** (Le & Woo, 2023), **CogVLM** (Wang et al., 2023a), **Instruct-BLIP** (Dai et al., 2023), and **InstructBLIP with LoRA tuning** (Hu et al., 2021). For the detailed explanation of the checkpoints we use for every baseline model, please refer to the supplementary materials.

Table 2: **Held-in and held-out deepfake detection accuracies**. Experiments are conducted on 2 real and 21 fake datasets, including 3 attacked ones. The accuracies of real and fake held-in datasets are marked by  grey background color , and those without background color are the accuracies of held-out datasets. The best and the 2nd-best performances are denoted in **bold** and underlined. `P`, `F`, `LoRA`, `Orig.` and `+LaMa` stand for different variants of each method, i.e., `P`: pretrained; `F`: directly-finetuned on AntifakePrompt's training dataset; `LoRA`: LoRA-finetuned on Antifake-Prompt's training dataset; `Orig.`: trained on original training dataset; `+LaMa`: trained on training dataset with additional LaMa-generated images.

| Dataset | Ricker2022 P | ResNet F | FatFormer P | Wang2020 P | Wang2020 F | DE-FAKE P | DE-FAKE F | DIRE P | DIRE F | LASTED P | LASTED F | QAD P | QAD F | CogVLM P | InstructBLIP P | InstructBLIP LoRA | AntifakePrompt Orig. | AntifakePrompt +LaMa |
|---|---|---|---|---|---|---|---|---|---|---|---|---|---|---|---|---|---|---|
| COCO | 95.60 | 99.43 | 97.40 | 96.87 | **99.97** | 85.97 | 83.30 | 81.77 | _99.93_ | 75.47 | 58.10 | 59.57 | 96.83 | 98.43 | 98.93 | 97.63 | 92.53 | 90.40 |
| Flickr | 95.80 | 99.23 | 98.13 | 96.67 | **100.00** | 90.67 | 84.38 | 77.53 | _99.93_ | 76.33 | 65.58 | 60.23 | 98.30 | 99.63 | 99.63 | 97.50 | 91.57 | 90.60 |
| SD2 | 81.10 | 2.50 | 16.83 | 0.17 | 5.23 | 97.10 | 88.07 | 3.83 | 30.47 | 58.69 | 52.53 | 51.00 | 10.67 | 52.47 | 40.27 | 89.57 | **98.33** | _97.97_ |
| SD3 | 88.40 | _99.83_ | 21.50 | 4.70 | 8.60 | 96.50 | 95.17 | 0.00 | 98.53 | 78.68 | 79.51 | 46.53 | **99.97** | 2.10 | 1.47 | 97.60 | 96.17 | 96.10 |
| SDXL | 81.10 | 0.50 | 30.39 | 0.17 | 1.53 | 90.50 | 72.17 | 18.17 | 19.73 | 51.33 | 77.65 | 41.60 | 9.87 | 32.57 | 23.07 | 96.47 | _99.17_ | **99.37** |
| IF | 92.65 | 4.40 | 27.73 | 19.17 | 4.93 | **99.20** | 95.20 | 6.93 | 63.17 | 57.99 | 55.63 | 59.07 | 15.17 | 29.03 | 20.63 | 87.90 | _97.10_ | 95.97 |
| DALLE-2 | 52.10 | 12.80 | 76.03 | 3.40 | 0.87 | 68.97 | 61.17 | 2.13 | 1.50 | 57.96 | 81.91 | 41.70 | 14.63 | 60.70 | 41.77 | **99.27** | 97.27 | _98.00_ |
| DALLE-3 | **95.20** | 2.10 | 43.97 | 18.17 | 3.20 | _89.00_ | 71.57 | 0.10 | 36.27 | 51.83 | 53.00 | 51.23 | 9.83 | 6.03 | 6.63 | 67.87 | 80.80 | 82.97 |
| playground v2.5 | 94.40 | 0.20 | 29.83 | 15.73 | 0.47 | 96.20 | 86.77 | 0.17 | 17.73 | 70.95 | 65.42 | 38.73 | 2.47 | 13.37 | 6.70 | 95.43 | _97.73_ | **98.13** |
| DiffusionDB | 81.20 | 4.69 | 60.50 | 9.03 | 4.50 | 80.80 | 78.10 | 2.53 | 16.40 | 86.48 | 67.42 | 52.07 | 12.07 | 6.05 | 53.00 | 85.40 | _98.47_ | **98.90** |
| SGXL | **100.00** | 1.63 | 97.73 | 79.30 | 2.13 | 56.90 | 50.20 | 45.27 | 9.50 | 64.39 | 65.59 | 46.40 | 4.20 | 60.40 | 69.53 | 91.20 | 99.03 | _99.37_ |
| GLIDE | 83.80 | 49.97 | 79.80 | 17.23 | 5.87 | 76.50 | 50.20 | 4.63 | 41.77 | 54.46 | 68.19 | 53.63 | 50.27 | 59.90 | 37.97 | 92.63 | _98.90_ | **99.70** |
| Stylization | 75.50 | 0.90 | 85.03 | 11.40 | 4.17 | 63.97 | 55.17 | 9.90 | 6.30 | 50.70 | 67.79 | 51.93 | 7.93 | 42.90 | 33.97 | 82.80 | _94.10_ | **95.77** |
| DF | 14.20 | 34.20 | 5.10 | 0.30 | 0.03 | 86.97 | 77.17 | 0.27 | 3.77 | 86.38 | 59.36 | _97.43_ | 22.73 | 13.80 | 13.83 | 67.43 | 95.03 | **98.40** |
| DFDC | 46.90 | 14.20 | 1.60 | 0.00 | 0.00 | 56.13 | 48.57 | 60.13 | 1.03 | 70.19 | 72.42 | 90.40 | 28.50 | 9.00 | 14.07 | 85.47 | _99.83_ | **99.93** |
| FF++ | 20.30 | 37.53 | 71.30 | 5.23 | 0.23 | 78.90 | 70.63 | 25.50 | 31.93 | 70.69 | 56.50 | 99.47 | 35.66 |  | 44.20 | 38.30 | 95.63 | _97.97_ |
| LaMa | 64.30 | 1.87 | _67.03_ | 7.53 | 0.07 | 13.03 | 23.00 | 13.23 | 19.47 | 60.53 | 97.67 | 42.03 | 3.80 | 5.20 | 10.90 | 42.73 | 39.40 | **55.80** |
| SD2IP | 59.10 | _99.76_ | 85.07 | 1.27 | 7.23 | 16.00 | 75.57 | 11.37 | 86.40 | 56.96 | **99.87** | 52.07 | 35.50 |  | 44.23 | _91.13_ | 80.80 | 89.03 |
| LIIF | 58.90 | 94.43 | 6.60 | 8.30 | 1.07 | 9.73 | 53.67 | 1.10 | 48.77 | 56.46 | 87.34 | 48.07 | 95.83 | 23.47 | _99.93_ | 84.63 | 98.50 | **99.97** |
| SD2SR | 73.90 | 97.79 | 84.03 | 1.40 | 0.13 | 29.70 | 96.67 | 2.77 | 27.20 | 59.59 | 99.73 | 47.50 | 8.63 | 55.06 | 69.10 | **99.90** | 99.43 | _99.80_ |
| Adver. | 8.50 | **99.36** | 2.53 | 3.97 | 0.03 | 60.40 | 82.77 | 1.60 | 3.93 | 59.03 | 63.06 | 49.23 | 6.37 | 7.23 | 5.50 | 51.77 | 87.10 | _97.83_ |
| Backdoor | 34.50 | _92.00_ | 15.70 | 15.47 | 0.00 | 22.23 | 89.37 | 1.93 | 2.77 | 52.63 | 77.35 | 38.37 | 15.73 | 2.83 | 3.17 | 41.10 | 86.20 | **96.83** |
| Data Poison. | 6.90 | **97.53** | 0.10 | 0.97 | 0.40 | 55.87 | 74.90 | 1.00 | 6.60 | 52.43 | 87.01 | 40.67 | 18.30 | 2.17 | 1.60 | 39.10 | 88.63 | _94.63_ |
| Average | 65.41 | 45.52 | 48.00 | 18.11 | 10.90 | 66.14 | 72.34 | 16.17 | 33.61 | 63.48 | 72.11 | 54.33 | 33.01 | 33.98 | 36.53 | 81.43 | _91.81_ | **94.50** |

## 4.2 Ablation Studies

**Pretrained CogVLM vs. Pretrained InstructBLIP.** As our proposed framework ideally is able to support any VLM, here we first take two representative VLMs (i.e., CogVLM and InstructBLIP, both pretrained) published recently to perform the pilot study. As shown in the columns marked as "CogVLM" and "InstructBLIP" with variant "P" in table 3, pretrained InstructBLIP demonstrates relatively average performance throughout all testing datasets, and reaches higher accuracies in inpainting and super-resolution datasets than pretrained CogVLM. This indicates that InstructBLIP relatively average ability to capture both coarse and fine artifacts of fake images, showing greater potential to detect fake images than CogVLM. Therefore, we select InstructBLIP as the backbone for our full model of AntifakePrompt in the following experiments.

**Position of $S_*$ in the tuned prompt.** We first investigate how the positioning of pseudo-word $S_*$ in the tuned prompt can influence our detector. Specifically, we compare 3 different positions of placing pseudo-word: replacing the word "real" in the prompt with pseudo-word, positioning the pseudo-word in the beginning of the prompt, or placing it at the end of the prompt. For simplicity, we refer to them as "replace", "prefix" and "postfix", respectively. As presented in Column 2 to 4 in table 1, although they all yield overall high accuracies, the "postfix" exhibits a slight advantage over the other alternatives. This suggests placing the pseudo-word at the end of prompt makes the best efforts among 3 different positions to enable deepfake detection, although the performance difference is not sensitive.

**Prompt tuning for the Q-former, LLM or both?** We then investigate the impact of prompt tuning by comparing the results of applying prompt tuning exclusively to Q-former, LLM or both modules. As shown in Column 5 to 7 in table 1, we observe that prompt tuning for both Q-former and LLM outperforms the other two alternatives in average accuracy. This implies that prompt tuning for both modules are benefical: tuned prompts to Q-former allow it to extract visual features from input image embeddings that are more conducive to differentiating between real and fake images. Tuned prompts to LLM can more precisely describe the idea of fake image detection for LLM, and thus, LLM is able to make more accurate decisions on differentiating real and fake images. Due to the improved visual features and improved instruction, the application of prompt tuning for both Q-former and LLM yields better performance.

**Number of training images.** We also study the effect of the number of real images in the training dataset. While fixing the number of fake images in training dataset to 60K, we gradually increase the

number of real images from 30K to 120K in the step of 30K, resulting in the total size of our training dataset ranging from 90K to 180K. As shown in Column 8 to 11 in table 1, while the accuracies on testing datasets of real images increase along with the increments of real images in training dataset, our detector suffers from a decrease in accuracies on fake testing datasets. Thus, we adopt the detector trained on 150K training dataset as our optimal model to achieve balanced accuracies on real and fake images.

To explore the limit of few-shot learning ability of our detector, we gradually reduce the numbers of both real images and fake images in training dataset to one tenth at each step until only 150 images remain in total. As shown in Column 12 to 14 in table 1, we found out that our detector still outperforms DE-FAKE in almost every testing datasets (except for SD2, SDXL and IF) when there are as few as 15K training samples, almost a quarter of DE-FAKE. Additionally, when we reduce our training dataset to only 1.5K images, only 0.2% of Wang2020 training dataset, our detector outperforms Wang2020 on every fake dataset, exhibiting only slightly lower accuracies on real datasets. These findings underscore the data efficiency of our detector in terms of training data size compared to eight baseline models.

## 4.3 Comparisons with SOTAs

**AntifakePrompt vs. Pretrained InstructBLIP.** As depicted in the columns marked as "Antifake-Prompt" with variant "`Orig.`" and "InstructBLIP" with variant "`P`" in table 3, our detector, trained only on images generated by SD2 and those from COCO dataset, exhibits excellent ($> 85\%$) performance on most held-in and held-out datasets. In contrast, InstructBLIP without prompt tuning demonstrates generally lower accuracies on most of the testing datasets except for COCO, Flickr30k, and LIIF. This implies that with the help of prompt tuning, our detector can better understand the deepfake detection task. Thus, our detector is capable of collecting more useful visual features from the input images, resulting in making more accurate decisions on distinguishing real images from fake ones.

**AntifakePrompt vs. Non-VLM Baselines.** As shown in the column marked as "Wang2020" with variant "`P`" in table 3, **Wang2020**, trained on images generated by ProGAN (Karras et al., 2017b) and ImageNet (Deng et al., 2009), exhibits satisfactory performance on StyleGAN-XL (denoted as SGXL) and yields excellent accuracies on COCO and Flickr30k, in contrast to our detector. However, we observe notable decreases in accuracy when it is tested on other held-out datasets. Since they consist of images generated by non-GAN-based models and these images do not share the same artifacts as those in ProGAN-generated images, the detector proposed by (Wang et al., 2020) is unable to differentiate such images by the traits learned from ProGAN-generated images. For **Ricker2022**, which uses the same settings as Wang2020 but trained on more datasets, shows acceptable performances on some diffusion- or GAN-generated datasets. Similar to Wang2020, Ricker2022 struggles to generalize beyond text-to-image models by the artifact learned from its training dataset. Regarding **DE-FAKE**, trained on images generated by SD and COCO, it shows impressive performance in real images and 3 diffusion-based models (i.e., SD2, SDXL, and IF) and Deeperforensics, as shown in the column marked as "DE-FAKE" with variant "`P`" in table 3. However, it struggles to achieve accuracies above 80% on other held-out datasets. Because the detector proposed in DE-FAKE uses a similar backbone as that in Wang2020, it suffers from similar accuracy drops when applying to images generated by unseen generative models. Even though it takes the corresponding captions into consideration, which allows it to detect the "informativeness" difference between synthesized and real images, it still fails to improve its performance on held-out datasets. This is because they are generated by newer models, and thus they can generate images as informative as real images.

As for **DIRE**, trained on images generated by ADM (Dhariwal & Nichol, 2021), the results are not as excellent as they demonstrated in their paper. The possible reason is that our testing dataset (e.g., SD2) comprises images generated by more advanced models than ADM, implying that the distribution of these images is closer to that of real images. Thus, the reconstruction errors of such images are smaller than those of the images generated by ADM, making DIRE harder to differentiate them from real images. Regarding **LASTED**, we observe performance drops on almost every dataset except for SGXL and 3 face swapping datasets. Since these datasets all employ GAN-based models or models with encoder/decoder structure during their generating process, LASTED can demonstrate relatively high accuracies due to the learned GAN-related artifacts from its training sets. As

for other datasets, LASTED exhibits performances similar to random guessing, underscoring the poor generalizability of LASTED. Although **QAD** indeed exhibits its excellent performance on 3 face swapping datasets, we observe that it shows relatively low accuracies when testing on other datasets. This indicates that QAD, trained only on 7 face swapping datasets, might not be able to generalize its detection ability to other types of fake images. Lastly, although **Fatformer** considers extra features learned in frequency domain, it fails to generalized it performances on models other than real datasets and SGXL, since the artifacts of diffusion-generated images in frequency domain are not as significant as those of GAN-based images as reported in (Ricker et al., 2022).

We also compare AntifakePrompt with 6 recent or classic non-VLM baselines trained on the training dataset of AntifakePrompt, as depicted in columns marked with variant `F` in table 3. Although these baseline methods demonstrate improved or even higher accuracies on some of the testing datasets, none of these baselines achieve comparable performance with respect to our AntifakePrompt, i.e., $> 85\%$ accuracies on most of held-in and held-out datasets.

Therefore, we can conclude that methods using different strategies or frameworks other than VLM, e.g. ResNet, generally demonstrate relatively low accuracies on almost every datasets comprising images generated by unseen models, implying their lacks of generalizability. In contrast, Antifake-Prompt can maintain its excellent performance on images generated by unseen models. To discuss the reason, the notable generalizability of LLM, brought by its large training corpus, gives the strong zero-shot ability of VLM, and thus enables AntifakePrompt to show its outstanding generalizability on unseen data. Also, AntifakePrompt consistently outperforms almost every baseline on the datasets generated by 3 attacking strategies. We conclude that AntifakePrompt is more sensitive to slight and malicious pixel perturbations than its opponents.

Additionally, to address the relatively lower performance observed on LaMa testing dataset, we conduct an experiment to include additional images generated by LaMa into our training dataset. Under this modified setting, as depicted in the columns marked as "AntifakePrompt" with variants "`Orig.`" and "`+LaMa`" in table 3, our detector gives generally comparable or even higher accuracies on almost every fake dataset compared to the original setting (the detector trained on 150K training dataset). However, these accuracy enhancements come at the cost of decreased accuracies on real datasets since our detector must now generalize to the inclusion of additional LaMa images in our training set.

**Finetuning InstructBLIP with LoRA.** Furthermore, we conduct extended experiments to compare between our prompt tuning and LoRA-based (Hu et al., 2021) InstructBLIP parameter finetuning. The results, as shown in columns marked as "AntifakePrompt" with variant "`Orig.`" and "Instruct-BLIP" with variant "`LoRA`" in table 3, reveal that while the detector finetuned with LoRA achieves comparable results in certain testing datasets, our detector consistently outperforms it in the three attack datasets. This underscores the sensitivity of our detector to such attack scenarios. Since additional LoRA matrices introduce relatively more learnable parameters into LLM (around 4M) than those introduced by prompt tuning (around 4K), it is more likely for LoRA-tuned InstructBLIP to overfit to artifacts of training datasets, resulting in accuracy drops when applied to fake datasets with different traits, namely three attack datasets.

## 5 CONCLUSION

In this paper, we propose a solution, **AntifakePrompt**, to the deepfake detection problem utilizing vision-language model to address the limitations of traditional deepfake detection methods when being applied on held-out dataset. We formulate the deepfake detection problem as a visual question answering problem, and apply soft prompt tuning on InstructBLIP. To evaluate the generalizability and effectiveness of AntifakePrompt, we carefully gather 23 datasets covering diverse real images and six commonly encountered synthetic scenarios, which can be used as benchmarks in deepfake detection for future studies. Empirical results demonstrate improved performance of our detector in both held-in and held-out testing datasets, which is trained solely on generated images using SD3 and real images from COCO datasets. Furthermore, in contrast to prior studies which require one to finetune/learn millions of parameters, our model only needs to tune 4864 trainable parameters, thus striking a better balance between the training cost and the effectiveness. Consequently, Antifake-Prompt provides a potent defense against the potential risks associated with the misuse of generative models, while demanding fewer training resources.

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

## A CHECKPOINTS OF EACH BASELINE

This section lists the checkpoint details of all the baselines mentioned in the paragraph "Baseline" of Section 3 in our main manuscript.

1. **Ricker2022 (Ricker et al., 2022)**: We use the checkpoint that was trained on images generated by 5 GANs and 5 DMs (i.e., ProGAN, StyleGAN, ProjectedGAN (Sauer et al., 2021), Diff-StyleGAN2 (Wang et al., 2022a), Diff-ProjectedGAN (Wang et al., 2022a), DDPM, IDDPM, ADM, PNDM (Liu et al., 2022) and LDM).

2. **ResNet** (He et al., 2016): We finetune the pretrained ResNet50 model, which is loaded from PyTorch Hub[4].

3. **Fatformer** (Liu et al., 2024): We use the checkpoint trained on ProGAN-generated 4-class data (car, cat, chair, horse) from (Wang et al., 2020), which can be found at their official GitHub page[5].

4. **Wang2020 (Wang et al., 2020)**: We use the detector checkpoint that is trained on dataset with images that are possibly Gaussian blurr- and JPEG-augmented, each with 10% probability.

5. **DE-FAKE (Sha et al., 2022)**: We use the checkpoint of the hybrid detector, which considers both the image and the corresponding prompts during detection.

6. **DIRE (Wang et al., 2023b)**: We use the checkpoint of detector trained on images from LSUN-Bedroom (LSUN B.) (Yu et al., 2015) and those generated by ADM.

7. **LASTED (Wu et al., 2023)**: We use the checkpoint of the detector trained on images from LSUN (Yu et al., 2015) and Danbooru (Dan, 2021), and those generated by ProGAN (Karras et al., 2017a) and SD1.5 (SD1).

8. **QAD (Le & Woo, 2023)**: We use the checkpoint of detector trained on 7 face swapping datasets (i.e., NeuralTextures (Thies et al., 2019), Deepfakes, Face2Face (Thies et al., 2016), FaceSwap (Thies et al., 2016), FaceShifter (Li et al., 2019), CelebDFv2 (Li et al., 2020) and FaceForensicsIntheWild).

9. **InstructBLIP (Dai et al., 2023)**: We use the pretrained weight provided by LAVIS and preset question prompt without prompt tuning.

10. **InstructBLIP (LoRA-tuned (Hu et al., 2021))**: We also use the pretrained weight provided by LAVIS and the preset question prompt, but apply LoRA tuning on LLM of InstructBLIP instead of prompt tuning.

11. **CogVLM (Wang et al., 2023a)**: We use the model weights finetuned on 8 open-sourced VQA datasets, including VQAv2 (Goyal et al., 2017), OKVQA (Marino et al., 2019), TextVQA (Singh et al., 2019), OCRVQA (Mishra et al., 2019), LLaVAInstruct (Liu et al., 2023b), ScienceQA (Saikh et al., 2022) and LRV-Instruction (Liu et al., 2023a), which can be found at the official GitHub page[6].

## B DATA GENERATION DETAILS

1. **Text-to-images generation**: We collected 3K prompts, half of which are sampled from the COCO ground truth captions and the other half from Flickr30k captions. These prompts are then input into six different generative models, i.e., SD2, SD3, SDXL, DeepFloyd IF, DALLE-2, DALLE-3 , Playground v2.5 , SGXL, and GLIDE, to generate the corresponding images. For DiffusionDB, we randomly sample 3000 images for testing set.

2. **Image stylization**: We begin by extracting Canny edge features from the 3000 test images in COCO dataset mentioned in the previous paragraph. Subsequently, we pass these Canny edge feature images, along with the corresponding prompts, into ControlNet to generate stylized images.

---

[4]https://pytorch.org/hub/pytorch_vision_resnet/

[5]https://github.com/Michel-liu/FatFormer?tab=readme-ov-file#model-zoo

[6]`https://github.com/THUDM/CogVLM/#model-checkpoints`

3. **Image inpainting**: We employ the same 3000 test images and resize them to make the shorter side of each image be 224, which matches the input size of InstructBLIP. Then, we randomly generate masks of three distinct thickness levels for these resized images using the scripts from the LaMa (Suvorov et al., 2022) GitHub[7]. With original images and the corresponding masks prepared, we utilize two different models, SD2-Inpainting (denoted as SD2IP) and LaMa, to inpaint images, respectively. The resizing step ensures that most of artifacts created during the inpainting process will be retained before being inputted to the detector.

4. **Super Resolution**: Out of the same reason in the inpainting, we apply the same resizing process to the same 3000 test images before downsizing them to one-forth of their original sizes. These low-resolution images are then passed into two different models, SD2-SuperResolution (denoted as SD2SR) and LIIF (Chen et al., 2021), to upsize back. A scaling factor of four is chosen, as only the ×4-upscaling weights for SD2 are publicly available.

5. **Face Swap**: Since face swapping is also one of the common means to generate fake images, we employ three large-scale face swapping video datasets, namely DeeperForensics (denoted as DF) (Jiang et al., 2020), DFDC (Dolhansky et al., 2019) and FaceForensics++ (denoted as FF++) (Rossler et al., 2019). From each of these datasets, we randomly extract 3, 3 and 4 frames from 1000, 1000 and 750 randomly selected videos, respectively. To ensure the extracted subset better represents the entire dataset, we randomly selected 125 videos from each of the 6 distinct categories within FF++. Following (Wang et al., 2020), we then apply Faced (Itzcovich, 2018) to crop out 3000 faces from the extracted frames of each dataset to ensure that complete facial features are present in every image.

6. **Image attacks**: We apply three common types of attacks to edit images and target at a traditional ResNet-50 classifier. The attack types include adversarial attack (Kim, 2020), backdoor attack (Li et al., 2021) and data poisoning attack (Geiping et al., 2020) (denoted as Adver., Backboor, and Data Poison., respectively). Default settings are employed for each attack. By testing our detector on these attacks, we can have a better understanding of its sensitivity against these slight and malicious image editing.

## C  TRAINING DATASETS AND TRAINABLE PARAMETERS COMPARISON

We list the size of training datasets and the number of trainable parameters of each baseline method along with those of AntifakePrompt for comparisons. Together with Table 1 in the main manuscript, we can conclude that AntifakePrompt demonstrates excellent performance at less number of training dataset and less trainable parameters among all other opponents.

Table 3: **Training datasets and trainable parameters.** The least number of training dataset/trainable parameters is denoted in **bold**. The 2nd-least number of training dataset/trainable parameters is denoted in underline.

| Methods | No. of training dataset | No. of trainable param. |
|---|---|---|
| Ricker2022 | 195K | 23.51M |
| Wang2020 | 720K | 23.51M |
| DE-FAKE | **40K** | 308.02M |
| DIRE | 200K | 23.51M |
| LASTED | 792K | 625.63M |
| QAD | 636K | **2.56K** |
| CogVLM | >1.5B | 6.5B |
| InstructBLIP | 15.183M | 188.84M |
| InstructBLIP (LoRA-tuned) | 150K | 4.19M |
| AntifakePrompt | 150K | 4.86K |

---

[7]https://github.com/advimman/lama/blob/main/bin/gen_mask_dataset.py

## D    SAMPLES FOR EACH DATASET

We provide some samples for each testing set and its correspong accuracy. Please refer to Figure 2 to Figure 24.

**MS-COCO**

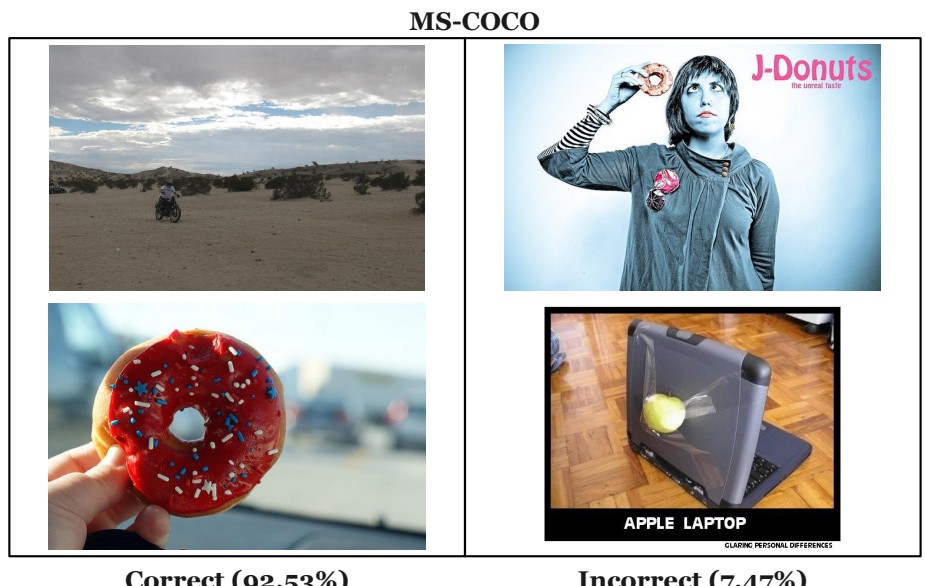

Correct (92.53%)              Incorrect (7.47%)

Figure 2: **Samples for MS-COCO.** 92.53% of images in the MS-COCO are correctly classified as real.

**Flickr30k**

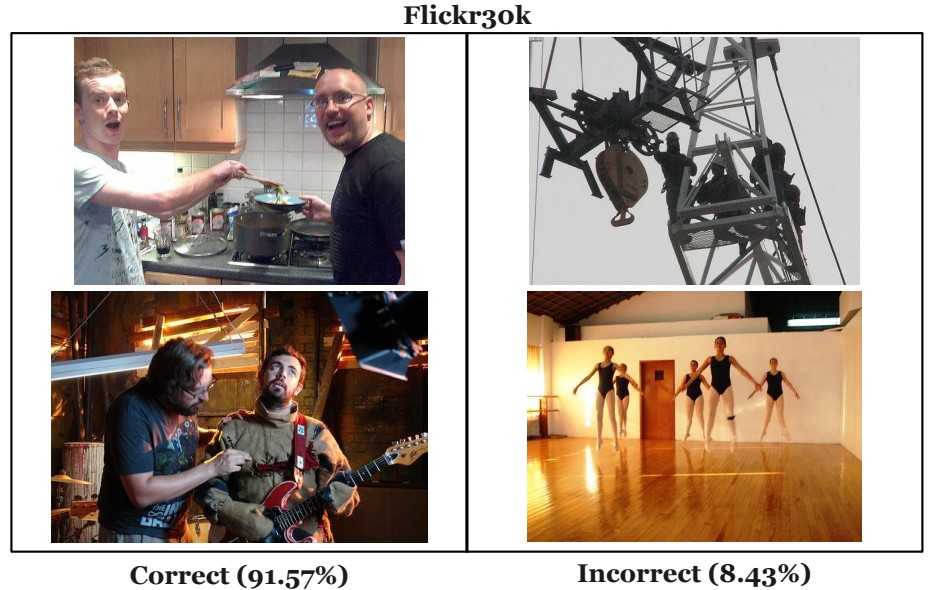

Correct (91.57%)              Incorrect (8.43%)

Figure 3: **Samples for Flickr30k.** 91.57% of images in the Flickr30k are correctly classified as real.

**SD2**

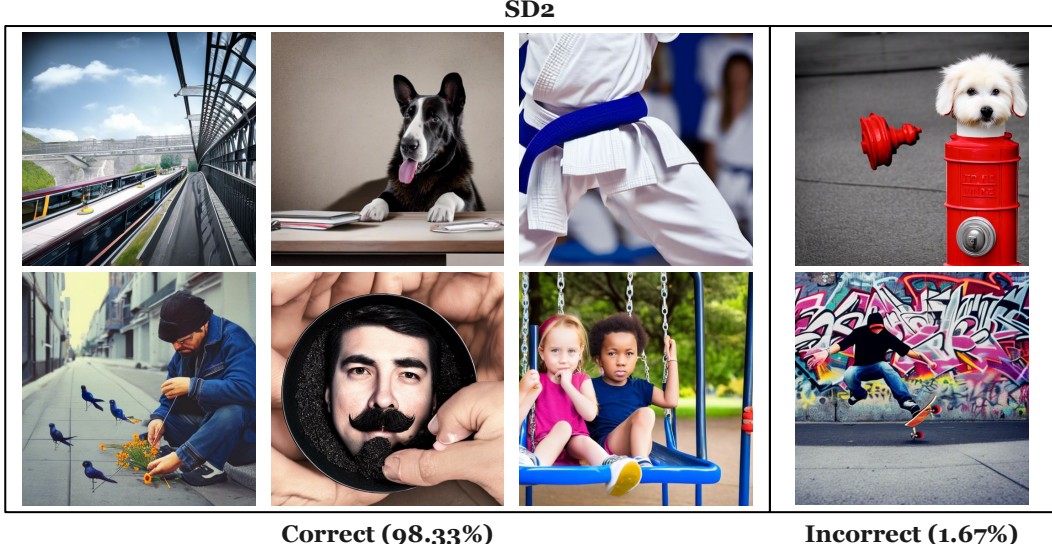

**Correct (98.33%)**          **Incorrect (1.67%)**

Figure 4: **Samples for SD2.** 98.33% of images generated by SD2 are correctly classified as fake.

**SD3**

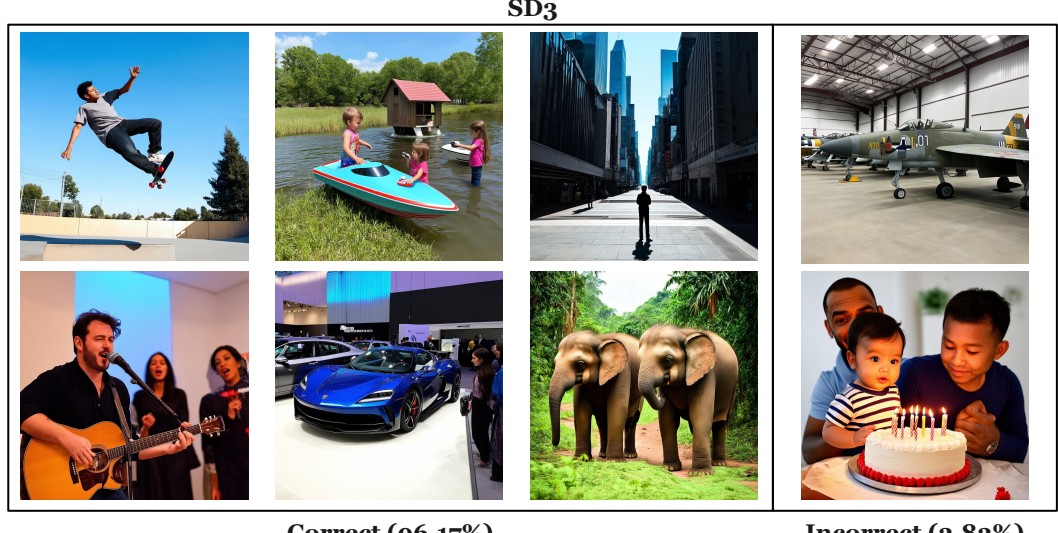

**Correct (96.17%)**          **Incorrect (3.83%)**

Figure 5: **Samples for SD3.** 96.17% of images generated by SD3 are correctly classified as fake.

**SDXL**

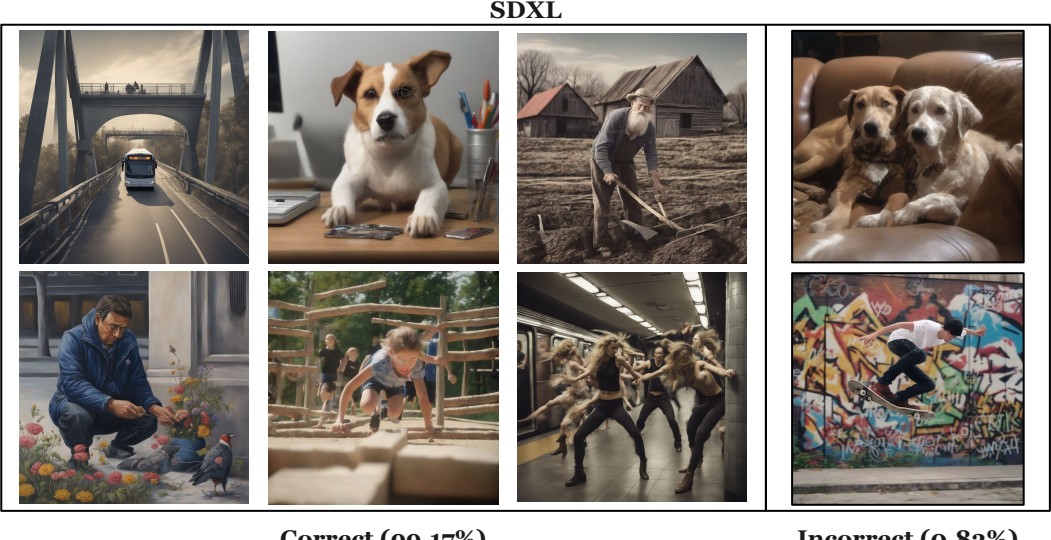

**Correct (99.17%)**                **Incorrect (0.83%)**

Figure 6: **Samples for SDXL.** 99.17% of images generated by SDXL are correctly classified as fake.

**IF**

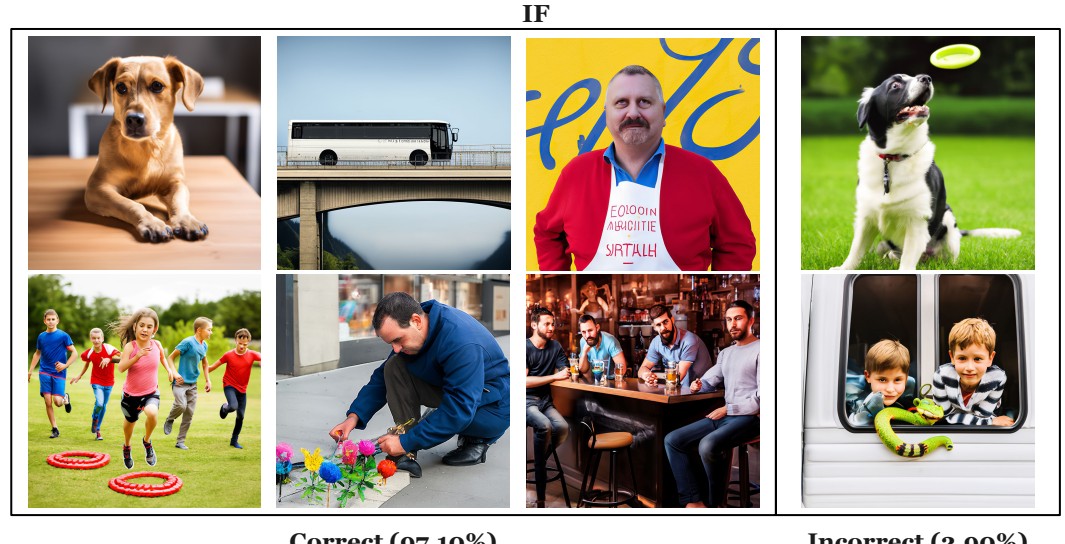

**Correct (97.10%)**                **Incorrect (2.90%)**

Figure 7: **Samples for IF.** 97.1% of images generated by IF are correctly classified as fake.

**DALLE-2**

Correct (97.27%)        Incorrect (2.73%)

Figure 8: **Samples for DALLE-2.** 97.27% of images generated by DALLE-2 are correctly classified as fake.

**DALLE-3**

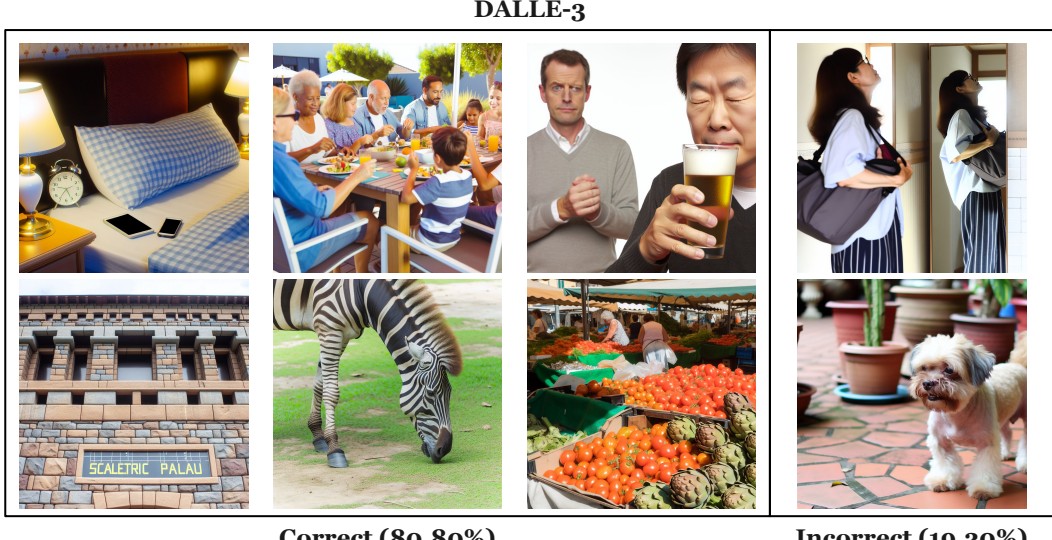

Correct (80.80%)        Incorrect (19.20%)

Figure 9: **Samples for DALLE-3.** 80.80% of images generated by DALLE-3 are correctly classified as fake.

**Playground v2.5**

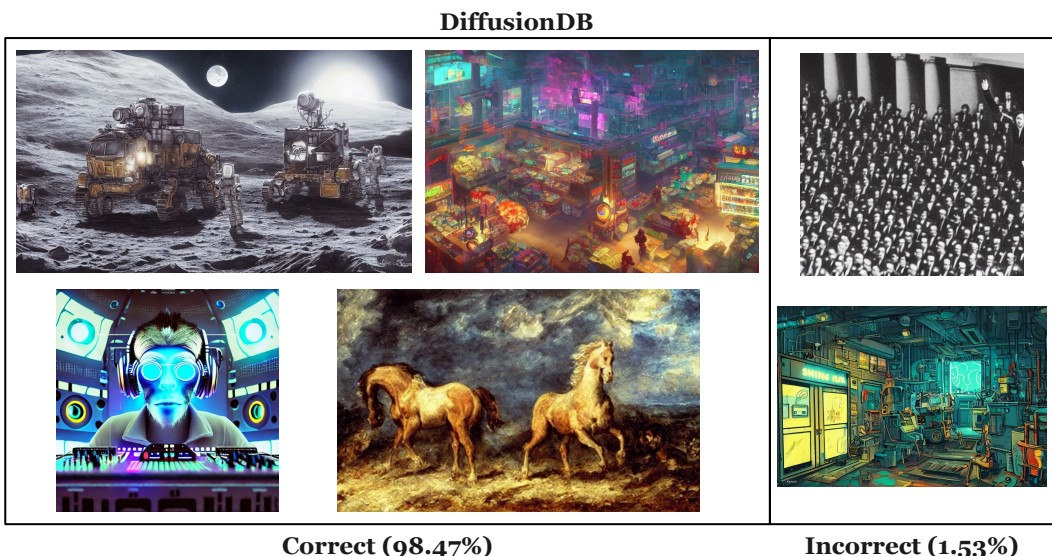

**Correct (97.73%)**     **Incorrect (2.27%)**

Figure 10: **Samples for Playground v2.5.** 97.73% of images generated by playground v2.5 are correctly classified as fake.

**DiffusionDB**

**Correct (98.47%)**     **Incorrect (1.53%)**

Figure 11: **Samples for DiffusionDB.** 98.47% of images generated by DiffusionDB are correctly classified as fake.

**SGXL**

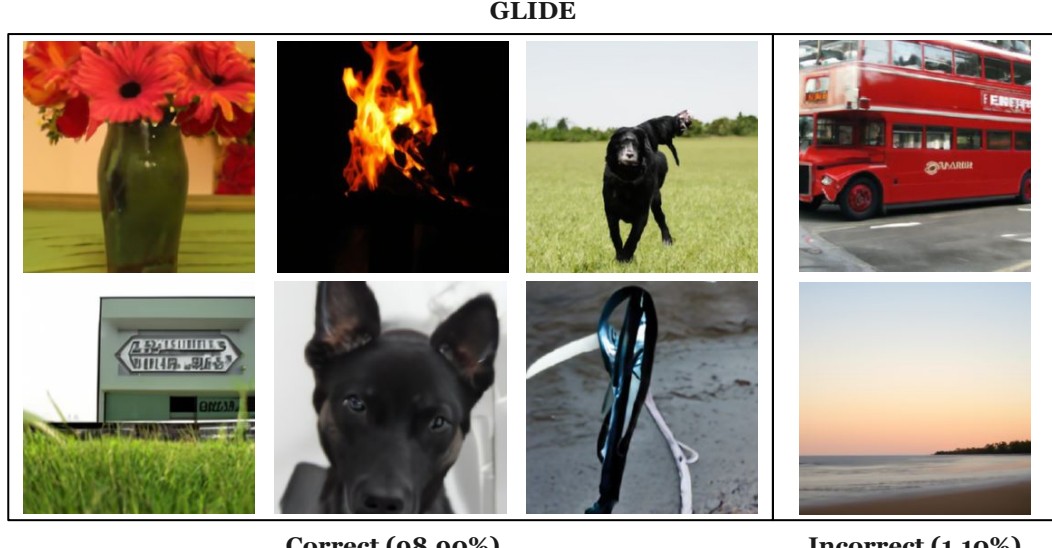

**Correct (99.03%)**                    **Incorrect (0.97%)**

Figure 12: **Samples for SGXL.** 99.03% of images generated by SGXL are correctly classified as fake.

**GLIDE**

**Correct (98.90%)**                    **Incorrect (1.10%)**

Figure 13: **Samples for GLIDE.** 98.9% of images generated by GLIDE are correctly classified as fake.

**Stylization**

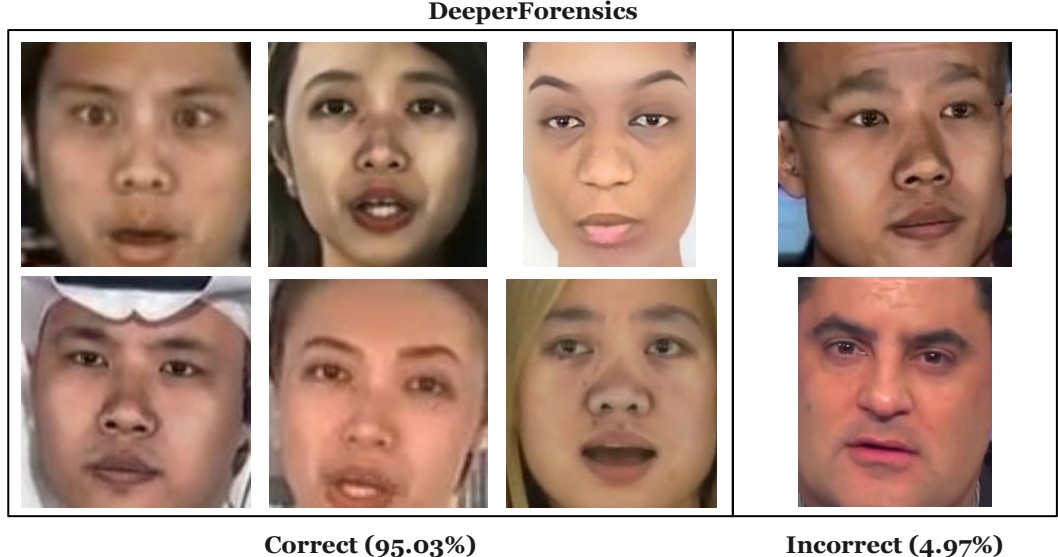

**Correct (94.10%)**                    **Incorrect (5.90%)**

Figure 14: **Samples for Stylization.** 94.1% of images generated by ControlNet are correctly classified as fake.

**DeeperForensics**

**Correct (95.03%)**                    **Incorrect (4.97%)**

Figure 15: **Samples for DeeperForensics.** 95.03% of images in DeeperForensics are correctly classified as fake.

**DFDC**

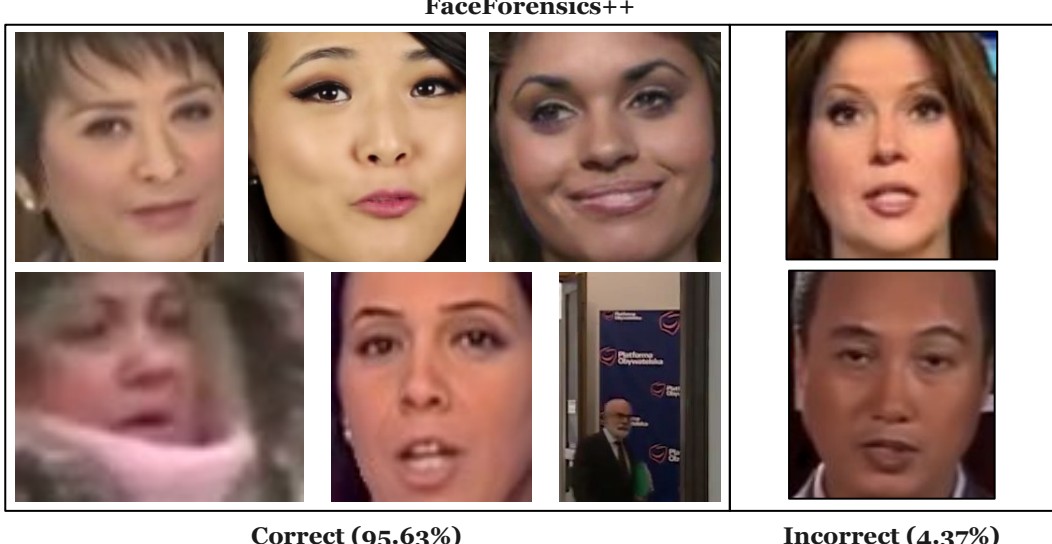

**Correct (99.83%)**          **Incorrect (0.17%)**

Figure 16: **Samples for DFDC.** 99.83% of images in DFDC are correctly classified as fake.

**FaceForensics++**

**Correct (95.63%)**          **Incorrect (4.37%)**

Figure 17: **Samples for FaceForensics++.** 95.63% of images in FaceForensics++ are correctly classified as fake.

**LaMa**

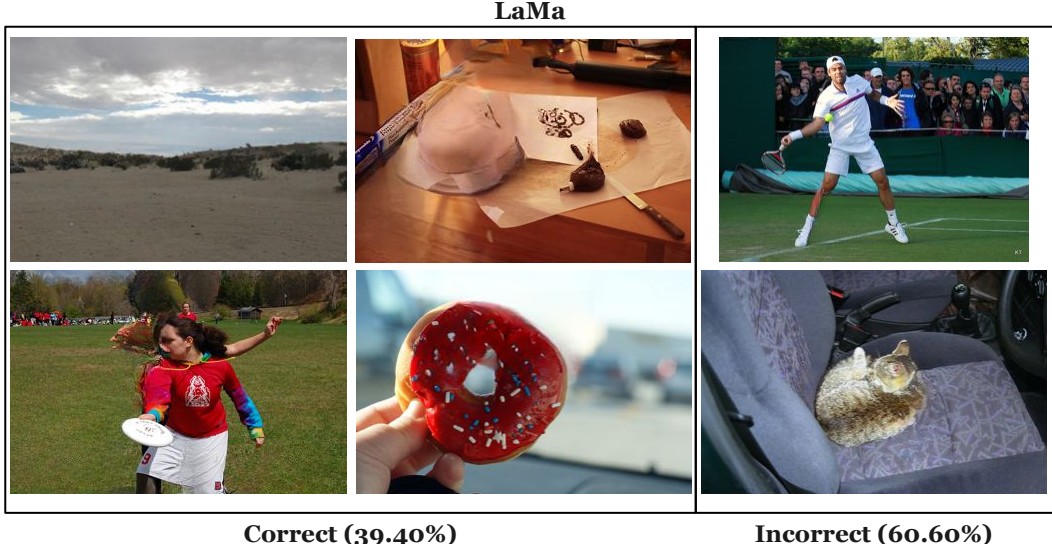

**Correct (39.40%)**                    **Incorrect (60.60%)**

Figure 18: **Samples for LaMa.** 39.4% of images generated by LaMa are correctly classified as fake.

**SD2-Inpainting**

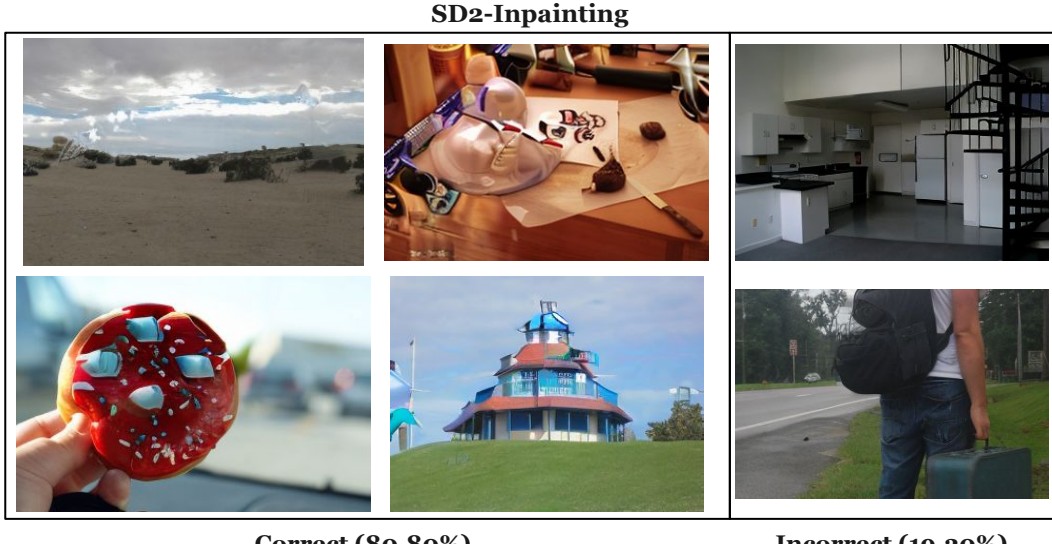

**Correct (80.80%)**                    **Incorrect (19.20%)**

Figure 19: **Samples for SD2-Inpainting).** 80.8% of images generated by SD2-Inpainting(SD2-IP) are correctly classified as fake.

**LIIF**

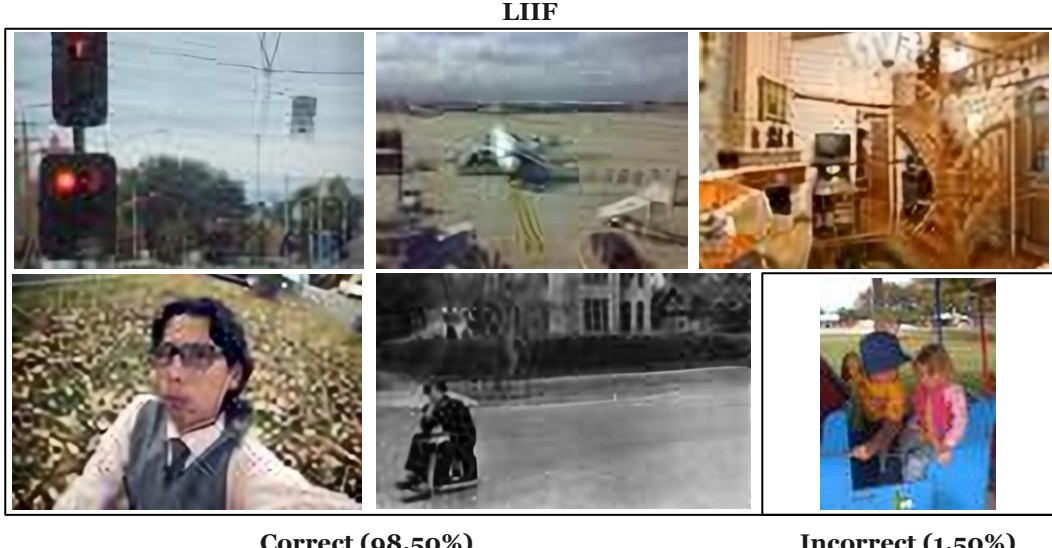

**Correct (98.50%)**             **Incorrect (1.50%)**

Figure 20: **Samples for LIIF.** 98.5% of images generated by LIIF are correctly classified as fake.

**SD2-SuperResolution**

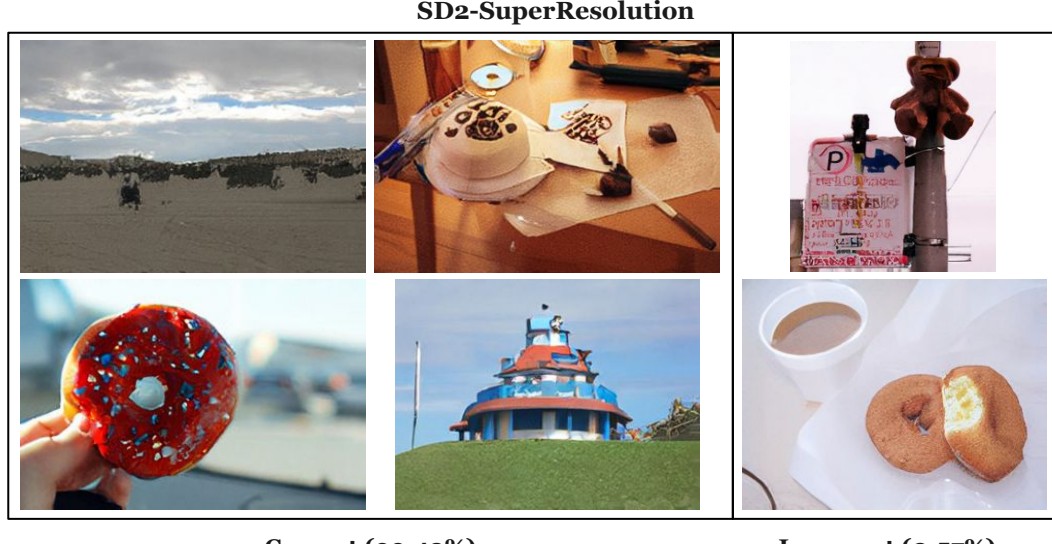

**Correct (99.43%)**             **Incorrect (0.57%)**

Figure 21: **Samples for SD2SR.** 99.43% of images generated by SD2-SuperResolution(SD2-SR) are correctly classified as fake.

**Adversarial Attack**

**Correct (87.10%)**    **Incorrect (12.90%)**

Figure 22: **Samples for adversarial attack.** 87.1% of images generated under adversarial attack are correctly classified as fake.

**Backdoor Attack**

**Correct (86.20%)**    **Incorrect (13.80%)**

Figure 23: **Samples for backdoor attack.** 86.2% of images generated under backdoor attack are correctly classified as fake.

**Data Poisoning Attack**

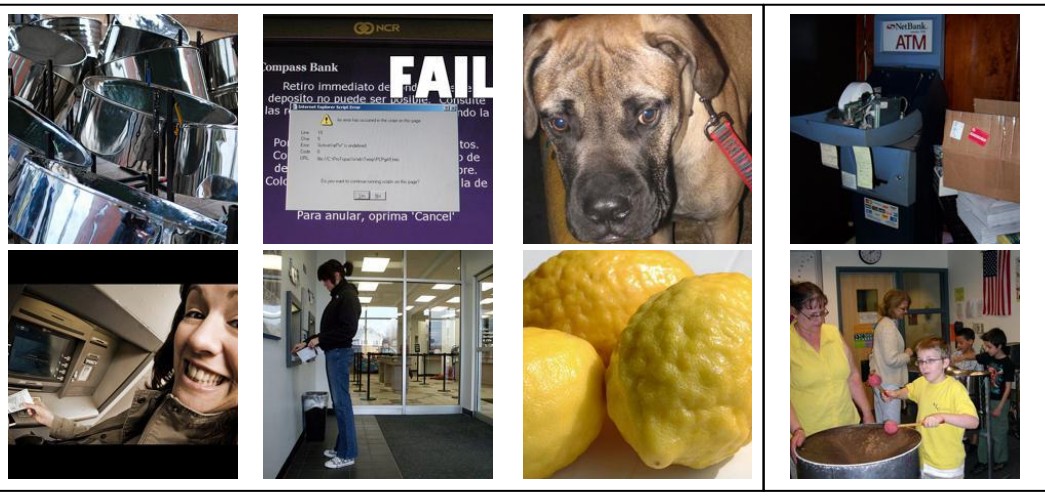

**Correct (88.63%)** **Incorrect (11.37%)**

Figure 24: **Samples for data poisoning attack.** 88.63% of images generated under data poisoning attack are correctly classified as fake.

