# OpenReview forum: "AntifakePrompt: Prompt-Tuned Vision-Language Models are Fake Image Detectors"
_ICLR.cc/2025/Workshop/BuildingTrust — BuildingTrust_

### Official Review · Reviewer_uNMu · 2025-02-24
***AntifakePrompt* is a deepfake detection method that uses vision-language models and prompt tuning to distinguish fake images across various generative models. This approach improves accuracy and generalizability, outperforming traditional methods. Despite its strengths, the study is limited by the use of diffusion-based fake image datasets, which could introduce bias, and lacks a detailed explanation of the impact of prompt tuning on performance improvements.**

**Rating:** 6
**Confidence:** 4

**Review:**

**Summary**

The paper introduces AntifakePrompt, a deepfake detection method leveraging vision-language models and prompt tuning. The approach frames deepfake detection as a visual question-answering task, enhancing accuracy by using pretrained VLMs like InstructBLIP. Through prompt tuning, the model achieves high performance on both seen and unseen deepfake data, improving detection capabilities with fewer training resources

**Strength**

1. It formulates deepfake detection as a visual question-answering task and leverages vision-language models with prompt tuning, which is an effective strategy for improving generalizability and accuracy.

2. The proposed method consistently outperforms traditional deepfake detection methods and state-of-the-art approaches across a wide range of datasets, including those from emerging generative models.

3. The model achieves high accuracy with significantly fewer training parameters and data, making it a cost-effective solution for deepfake detection compared to other models that require extensive fine-tuning.

**Weakness**

1. This study evaluates fake image datasets generated exclusively by diffusion-based models, which may introduce bias.
2. The paper lacks a comprehensive analysis of why tuning a single word in the prompt within the VLM yields significant performance improvements.

---

### Decision · Program_Chairs · 2025-03-04

Accept